# Development of a Particle Filter-Based Path Tracking Algorithm of Autonomous Trucks with a Single Steering and Driving Module Using a Monocular Camera

**DOI:** 10.3390/s23073650

**Published:** 2023-03-31

**Authors:** Sehwan Kim, Munjung Jang, Hanbyeol La, Kwangseok Oh

**Affiliations:** School of ICT, Robotics & Mechanical Engineering, Hankyong National University, Anseong-si 17579, Republic of Korea; ksh49@hknu.ac.kr (S.K.); munjung0831@hknu.ac.kr (M.J.); byeol0515@hknu.ac.kr (H.L.)

**Keywords:** particle filter, monocular camera, path tracking, autonomous trucks, single steering and driving module, linear quadratic regulator

## Abstract

Recently, in various fields, research into the path tracking of autonomous vehicles and automated guided vehicles has been conducted to improve worker safety, convenience, and work efficiency. For path tracking of various systems applied to autonomous driving technology, it is necessary to recognize the surrounding environment, determine technology accordingly, and develop control methods. Various sensors and artificial-intelligence-based perception methods have limitations in that they must learn a large amount of data. Therefore, a particle-filter-based path tracking algorithm using a monocular camera was used for the recognition of target RGB. The path tracking errors were calculated and a linear-quadratic-regulator-based desired steering angle were derived. The autonomous trucks were steered and driven using a pulse-width-modulation-based steering and driving motor. Based on an autonomous truck with a single steering and driving module, it was verified that the path tracking could be used in three evaluation scenarios. To compare the LQR-based path tracking control performance proposed in this paper, an elliptical path tracking scenario using a conventional sliding mode control with robust control performance was performed. The results show that the RMS of the lateral preview error of the SMC was approximately 18% larger than that of the LQR-based method.

## 1. Introduction

Recently, in various fields, such as logistics, agriculture, and transportation, autonomous driving technology has been used to increase the safety, convenience, and efficiency of workers. In particular, in logistics or factory industrial sites, robots and automated guided vehicles (AGVs) with autonomous driving technology are used to move and classify heavy goods. The AGV is responsible for loading goods, tracking workers, or driving to the target point. At work sites, early infrastructure was composed of magnetic, barcode, beacon, and color bands, and the surrounding environment was perceived using various sensors, such as cameras, lidar, and radar. Based on the surrounding environment information obtained through sensors, various control methods, such as artificial intelligence and optimization control, are used to determine and control the target path of the autonomous driving system. Systems with autonomous driving technology used in various fields require a control method to firmly recognize and decide the surrounding environment and a given path, and to track the path. To track the path of systems to which this autonomous driving technology is applied, various recognition, decision, and control methods are being researched and developed.

### Literature Review

Tang et al., proposed a path tracking controller that combines a kinematic model predictive controller and a proportional–integral–derivative (PID) controller to reduce the uncertainty of the model and compensate for the side slip in the high-speed driving situations of autonomous vehicles [1]. Lee et al., proposed an adaptive linear quadratic Gaussian control scheme to reduce uncertainties owing to various driving environment changes and sensor noise [2]. Taghavifar et al., proposed an exponential sliding mode fuzzy type neural network control method to reduce the nonlinearity of autonomous vehicle systems and the uncertainty of the mathematical model, and to improve the path tracking and lane-keeping performance [3]. Awad et al., proposed a linear model predictive controller with a Laguerre network for longitudinal and lateral control of path tracking by linearizing a fuzzy-model-based nonlinear system [4]. Mishra et al., derived the relative pixel density around the path while minimizing the uncertainty of illumination, occlusion, and observation images using a fuzzy-system-based vision sensor. The pulse-width modulation (PWM) control method of an autonomous vehicle motor for path tracking was proposed using a neural network with the derived relative pixel density as the input [5]. Hang et al., proposed a path tracking controller by deriving the optimal four-wheel steering angle by integrating the linear quadratic regulator (LQR) with the feedforward controller based on the linear parameter variable system model of an autonomous vehicle [6]. Chen et al., proposed an active disturbance rejection control backstepping control method to track the path of an AGV whose linear and angular velocities are uncertain. Because both the position and angle of the AGV are affected by uncertainty, two extended state observer-based real-time total disturbances were estimated [7]. Enrique et al., derived a reinforcement-learning-based desired angular velocity for the path tracking of an AGV and proposed a control method through PI regulator-based speed control. When only the PID controller was used to change the friction coefficient and in the complicated path driving scenario, a reasonable performance was confirmed [8]. Chen et al., proposed a navigation algorithm using a neural network by collecting human decision-making process data for autonomous mobile robots. Training data were collected on various scenarios with randomly generated obstacles. The proposed neutral network algorithm was trained using a supervisory learning method without global environmental information. It addressed computation load issue and achieved high estimation accuracy [9].

For autonomous driving, paths and obstacles should be detected by measuring the surrounding environment and system information based on various sensors. A number of studies have used artificial intelligence methods to reduce the noise of sensors and detect paths. In addition, adaptive methods are used to reduce the uncertainty between the mathematical model and the actual system, such as autonomous vehicles and AGV.

A particle filter generates weighted particles to approximate posterior probability density function. It can be employed for nonlinear, non-Gaussian noise systems and may not require mathematical models. Therefore, it is used to design controllers by estimating system parameters or states [10,11,12]. However, in most related works, additional experimentation has been required for the verification and extension of the proposed algorithm in various plants and scenarios. Conducting additional experiments is challenging because of the high cost and time requirements. Therefore, an optimal algorithm is required to ensure usability and versatility for various environmental applications. In recent years, particle filters have been used in various areas, not only to track targets [13,14], but also in motion planning [15,16] for autonomous systems. It can be applied to segmentation problem using particle filters [17]. In this paper, a particle filter is applied to a design tracking algorithm for an autonomous truck using a monocular camera without additional environmental information and system models.

Liang et al., proposed a decoupling and a side-slip phase-plane-based control method to improve the lateral stability of four-wheel independent steering and autonomous vehicles under high-speed conditions [18]. Song et al., used an unscented Kalman filter (UKF) observer to estimate the longitudinal speed, lateral tire force, side slip angle, desired steering angle, and tire force for improving the side surface stability of a four-wheel independent steering vehicle, which were optimized by using an model predictive control (MPC) based on the estimated value [19]. Cheng et al., proposed an MPC that adjusted the weighting of the cost function to provide the necessary yaw moment to maintain the lateral stability of the vehicle in a steeply decelerated situation. Using the proposed controller, the performance was verified on a test bench consisting of wheel brakes, a yaw rate sensor, and wheel speed simulation modules [20]. Hang et al., proposed a tube-MPC-based control method that considers lateral stability constraints and path tracking error constraints to improve vehicle handling stability and path tracking performance [21]. Wang et al., proposed a path-following control method for an Ackerman-steered vehicle using a fuzzy rule-based weight control MPC [22]. Pereira et al., proposed an MPC control method based on mapping using two Gaussian function approximations to solve the inaccuracy problem of the kinematic bicycle model. The proposed controller was verified through a simulation and then evaluated in a real heavy-duty vehicle [23]. Liu et al., proposed an MPC algorithm with dynamic constraints by combining feedforward control as a method of tracking the path of a four-wheeled independent steering robot composed of a steering and driving motor, considering the operation delay of the steering motor [24]. Jeong et al., proposed an algorithm that tracks paths based on yaw rates and generates yaw moments to distribute them to four wheels based on weighted pseudo-inverse-based control allocation (WPCA) for improved path tracking performance in four-wheel independent steering and driving systems [25].

Methods of using yaw rates and adjusting weights were used to control the steering angle in path tracking. Most studies have controlled based on the lateral preview error and yaw angle error to converge the tracking path of an autonomous vehicle to the reference path. Normal MPC may have adverse affects on the control performance owing to the uncertainty between an actual system and a mathematical model. In addition, it has a large computational load. Therefore, methodologies that are robust and can reduce the computational load have been proposed for applications in real systems.

In previous studies, it was confirmed that research into autonomous vehicles and AGVs is being conducted to increase worker safety, convenience, and work efficiency. To track the path of various systems by applying autonomous driving technology, the surrounding environment should be perceived and the technology must be decided and controlled accordingly. For the path tracking of an autonomous driving system, various sensors and artificial intelligence methods are used to detect paths and obstacles, and adaptive theory or artificial-intelligence-based path tracking control methods are used. However, artificial intelligence requires a large amount of training data and image processing for path and obstacles detection. It was recently confirmed that particle filters can be used in object tracking and path planning methods as an advantage that can be applied to nonlinear systems without mathematical models. Therefore, in this paper, we propose a particle filter that does not require image processing for path tracking using RGB information from the obtained image. In addition, we propose a path tracking control algorithm using a simplified error equation and monocular-camera-based path error derived from the particle filter.

The remainder of this paper is organized as follows. Section 2 presents a particle-filter-based path tracking algorithm for an autonomous truck. Section 3 presents the results of performance evaluation under various scenarios. Section 4 concludes the paper with a discussion of the limitations of the current work and future research.

## 2. Particle-Filter-Based Path Tracking Algorithm

Figure 1 shows a model schematic of the path tracking algorithm of the overall autonomous trucks and is classified into the perception, decision, and control stages. In the perception stage, the path error was derived by using clustered particle points generated from the particle filter based on the defined target RGB value. In the decision stage, a decision on the driving and braking of the autonomous truck was made using the RGB distance between the target RGB and the RGB of the clustered point. The LQR-based desired steering angle was derived using the derived path error. In the control stage, the current steering angle was calculated by measuring the voltage of VR. For steering control, the PWM and pulse signal were applied to the step motor so that the calculated current steering angle converged to the desired steering angle. For driving control, the PWM signal was applied to the in-wheel motor based on the flags for driving and braking determined by the RGB distance error.

In Section 2.1, a particle-filter-based target RGB of a path using a camera is shown, and the path error derivation method is described. Section 2.2 describes the method of determining the driving and braking and the method for the desired steering angle of an autonomous truck based on path error, and Section 2.3 describes autonomous trucks with modular single steering and driving system.

### 2.1. Particle-Filter-Based Path Tracking Error Derivation Using Monocular Camera

Figure 2 shows a block diagram of the method used for tracking the target RGB and deriving path errors based on a particle filter using a monocular camera.

The particle filter algorithm for tracking the RGB of the target path comprises particle generation, particle update, likelihood calculation, and particle resampling. The RGB values are acquired from images within a monocular camera, and the positions and normally distributed velocity, with a mean of zero and a variance of 1, are randomly generated within the designed region of interest (ROI). The generated particles predict the movement position and velocity of the particles in the current state through the update process, as shown in Equation (1).
(1)$$\left[\begin{array}{c} x_{k}\\ y_{k}\\ {\dot{x}}_{k}\\ {\dot{y}}_{k} \end{array}\right]=\left[\begin{array}{cccc} 1 & 0 & 1 & 0\\ 0 & 1 & 0 & 1\\ 0 & 0 & 1 & 0\\ 0 & 0 & 0 & 1 \end{array}\right]\left[\begin{array}{c} x_{k-1}\\ y_{k-1}\\ {\dot{x}}_{k-1}\\ {\dot{y}}_{k-1} \end{array}\right]+\left[\begin{array}{c} \sigma_{pos}\gamma_{x}\\ \sigma_{pos}\gamma_{y}\\ \sigma_{vel}\gamma_{\dot{x}}\\ \sigma_{vel}\gamma_{\dot{y}} \end{array}\right]$$

As shown in Equation (1), the state variable is estimated at the current time (k) by adding the value obtained by multiplying the variable of the previous time (k−1) by the standard deviation of the position and velocity (σ) and the disturbance (γ).

In Equation (1), x and y are the positions of the particles; $\dot{x}$ and $\dot{y}$ are the velocities of the particles; $\sigma_{pos}$ and $\sigma_{vel}$ are the position and velocity of the particles, respectively; and $\gamma_{x}$, $\gamma_{y}$, $\gamma_{\dot{x}}$, and $\gamma_{\dot{y}}$ are the disturbances of randomly generated particles of position and velocity, respectively. In the likelihood calculation process, the Euclidean norm is derived as shown in Equation (2) using the target RGB and the RGB of the particles at the current location point, and the log likelihood is calculated as shown in Equation (3).
(2)$$d_{RGB}=\sqrt{{\left(r_{c}-r_{t}\right)}^{2}+{\left(g_{c}-g_{t}\right)}^{2}+{\left(b_{c}-b_{t}\right)}^{2}}$$
(3)$$L_{log}=\log\left(\frac{1}{\sqrt{2\pi }\sigma }e^{-\frac{d_{RGB}^{2}}{2\sigma^{2}}}\right)$$

Multinomial resampling methodology was used for the particle resampling process, and the maximum value of log likelihood was subtracted from the log likelihood derived from Equation (4) to make it all negative. The value of the difference was applied to the exponent of the natural constant to normalize the maximum value of likelihood to 1.
(4)$$L=e^{\left(L_{log}-\max\left(L_{log}\right)\right)}$$
(5)$$w_{t}=\overset{t}{\underset{i=t}{\sum }}L_{t}/\sum L$$

The weight $w_{t}$ is calculated as shown in Equation (5) using the normalized L and cumulative sum, and the low-weight particles to be derived are replaced with high-weight particles. When the particles move to the position most similar to the target RGB, the clustered point is derived through K-means clustering, and the clustered point is derived for each step by repeating the processes of particle update, likelihood calculation, particle resampling, and clustering of particles. By applying the ROI, the particles were created within the ROI and the desired preview distance was determined automatically. In addition, the ROI was applied for the particle generation and likelihood calculation steps to reduce target RGB tracking error by preventing the particles from distributing to points other than the target path. Figure 3 shows the concept of conversion of pixel to actual distance in image.
(6)$$Y=\left(0.0001Y_{pixel}{}^{2}-0.1786Y_{pixel}+109.09\right)/100$$
(7)$$X=\left(\left(-0.0001Y_{pixel}+0.0976\right)\times pixels\right)/100$$

To derive the path error between the target path and current camera position based on the derived clustered point, it is necessary to convert the pixels in the camera image into actual distances. To convert to actual distance, 15 cm rulers were placed at 20 cm intervals, and the actual distance according to the pixels were measured through an experiment. Figure 3 shows a measured image for actual distance conversion using a monocular camera with 1280×720 resolution used in the actual experiment. The actual distance according to the pixel position of the image acquired in the mounted camera was measured based on the camera in the autonomous trucks. As shown in Equations (6) and (7) to converted the actual distance based on the autonomous truck coordinates according to the pixels in the ROI. In Equations (6) and (7), $X_{pixel}$ and $Y_{pixel}$ are the coordinate points of the camera image width and height in pixels, and X and Y are the coordinates based on autonomous trucks and are the actual distances in the lateral and longitudinal directions within the camera image, respectively. The green dotted line represents the designed ROI.

As shown in Figure 4, the actual preview distance is derived as shown in Equation (8) using the preview point ($x_{p},L_{p}$) located at the center of the width of the camera, which changes according to the coordinate point of the pixel-based derived clustered point ($x_{1},y_{1}$). Equations (9) and (10) are the lateral preview error ($e_{y}$) and yaw angle error ($e_{\psi }$) derived based on pixels using the preview point.
(8)$$L_{p}=0.0001y_{1}^{2}-0.1786y_{1}+109.09$$
(9)$$e_{y}=\left(\left(-0.0001y_{1}+0.0976\right)\times \left(x_{p}-x_{1}\right)\right)/100$$
(10)$$e_{\psi }=\tan^{-1}\left(\frac{e_{y}}{L_{p}}\right)$$

In this study, we considered a clustered point as a tracking object and simplified it using a path tracking algorithm. Therefore, the lateral preview error and yaw angle error were derived based on pixels, as shown in Equations (9) and (10).

### 2.2. Steering and Driving Control Algorithm for Path Tracking

Figure 5 shows a block diagram of the steering control algorithm for path tracking. The desired steering angle based on the LQR was derived using path errors derived from the pixel data in the obtained image. For steering control, the steering angle error was calculated using the derived desired steering angle and the map-based current steering angle. A rotational flag that represents the step motor rotational direction using binary numbers such as 1 and 0 was generated using the calculated steering angle error. If the steering angle error is a positive, a counterclockwise flag is defined as 1 so that the step motor rotates counterclockwise until the steering error converges to zero. Conversely, if the steering angle error is a negative, a clockwise flag is defined as 1 so that the step motor rotates clockwise until the steering angle error converges to zero. The rotational flag was used to determine the rotational direction of the step motor and the step motor was controlled using the PWM signal based on the flag.

To derive the desired steering angle, the error equation was derived as Equations (11) and (12) based on the Ackerman geometry model.
(11)$$\left[\begin{array}{c} {\dot{e}}_{y}\\ {\dot{e}}_{\psi } \end{array}\right]=\left[\begin{array}{cc} 0 & v\\ 0 & 0 \end{array}\right]\left[\begin{array}{c} e_{y}\\ e_{\psi } \end{array}\right]+\left[\begin{array}{c} 0\\ \frac{v}{L} \end{array}\right]\delta_{des}+\left[\begin{array}{c} 0\\ -1 \end{array}\right]{\dot{\psi }}_{d}$$
(12)$$\dot{e}=Ae+B\delta_{des}$$

In Equation (11), the yaw rate is considered to be zero, and in Equation (12), A is the system matrix, B is the input matrix, and the input is the desired steering angle. Equation (13) is a feedback control input for minimizing the error states, which can be substituted into Equation (12) and represented as Equation (14):(13)$$\delta_{des}=-Ke$$
(14)$$\dot{e}=\left(A-BK\right)e$$
where K is the feedback gain matrix, and the LQR cost function and feedback gain matrix derivation process are shown in Equations (15)–(17).
(15)$$J=\int_{0}^{\infty }\left(e^{T}Qe+u^{T}Ru\right)dt$$
(16)$$PA+A^{T}P-PBR^{-1}B^{T}P+Q=0$$
(17)$$K=-R^{-1}BP$$

Equation (15) is the cost function of the LQR, and the feedback gain matrix is derived as shown in Equation (17) by deriving P, a positive definite matrix based on the Riccati equation (Equation (16)). Based on this, the desired steering angle, which is a control input that minimizes the cost function, was derived, as shown in Equation (18), where Q is the error state weighting matrix and R is the control input weighting matrix.
(18)$$\delta_{des}=Ke=k_{1}e_{y}+k_{2}e_{\psi }$$

The current steering angle is required to track the desired steering angle derived from the LQR. The current steering angle was derived using a map-based variable resistor (VR), as shown in Equations (19) and (20). The VR used had a voltage of 0 to 5 V, and by converting it into an analog signal of 0 to 1023, the current steering angle according to the voltage of VR within the designed minimum and maximum steering angle range was derived.
(19)$$A_{str}=V\times 1023/5$$
(20)$$\delta_{cur}=0.2685A_{str}-138.0615$$

The steering angle error is defined as shown in Equation (21) using the desired steering angle derived based on the LQR and the current steering angle derived based on the map. The current steering angle is driven based on the rotational direction and PWM of the pulse-signal-based step motor, as shown in Equations (22) and (23), to track the desired steering angle.
(21)$$e_{str}=\delta_{des}-\delta_{cur}$$
(22)$$Flag_{ccw}=\left\{\begin{array}{c} 0,\;\;e_{str}<0\\ 1,\;\;e_{str}\geq 0 \end{array}\right.$$
(23)$$Flag_{cw}=\left\{\begin{array}{c} 0,\;\;e_{str}\geq 0\\ 1,\;\;e_{str}<0 \end{array}\right.$$

When the steering error is positive, the PWM is applied to the step motor until the steering error converges to zero along with a counterclockwise pulse signal. Conversely, when the steering angle error has a negative sign, a clockwise pulse signal and PWM are applied to the step motor until the steering error converges to zero.
(24)$$e_{RGB}=\sqrt{{\left(r_{t}-r_{Clustered}\right)}^{2}+{\left(g_{t}-g_{Clustered}\right)}^{2}+{\left(b_{t}-b_{Clustered}\right)}^{2}}$$
(25)$$Flag_{dri}=\left\{\begin{array}{c} 0,\;\;e_{RGB}<d_{th}\\ 1,\;\;e_{RGB}\geq d_{th} \end{array}\right.$$

When the particles deviate from the target path, the error between the RGB data of the clustered point and target RGB data increases. Driving and braking were determined using the Euclidean norm of the RGB data of the clustered point and the error of the target RGB data, as shown in Equation (24). As shown in Equation (25), the PWM-based in-wheel motor was driven with a driving flag of zero when it was located within the designed error threshold range; when the error exceeded the threshold, the driving flag was changed to 1 and a braking signal was applied to the in-wheel motor. In Equations (22), (23) and (25), $Flag_{ccw},\;Flag_{cw}$, and $Flag_{dri}$ are dimensionless indices that have the values of 0 and 1.

### 2.3. Autonomous Truck with Single Driving Module

Figure 6 shows a block diagram of an autonomous truck with a single steering and driving module. The autonomous trucks developed in this study consist of a PC, a VR, a single monocular camera, and two micro control units (MCUs).

The PC implemented an RGB data-based particle filter algorithm obtained from a monocular camera and measured the voltage of VR to derive the steering angle error for controlling the step motor and error of the RGB data for controlling the in-wheel motor using MATLAB software. The PWM and braking signals were applied to the in-wheel motor through MCU-1 according to the determined driving flag based on the error of RGB data, and the steering rotation direction flag derived according to the steering error was transmitted to MCU-2. MCU-2 was driven based on pulse signal rotation direction switching and PWM-based operation of the step motor according to the steering rotation direction flag received through MCU-1.

The next section provides the performance evaluation results under various evaluation scenarios (ellipse path, s-curved path, and lane-change path tracking).

## 3. Performance Evaluation

Figure 7 shows the concept and an actual image of an autonomous trucks with a single steering and driving module. It consists of two 24 V batteries and one 6 V battery to supply power to the steering and driving module and the MCUs of the autonomous trucks.

Table 1 lists the specifications of autonomous trucks and mounted monocular cameras. The monocular camera was mounted at a height of 0.6 [m] from the ground and tilted downward by 48°. Table 2 lists the design parameters of the particle filter and control variables of the LQR, where $h_{pixel}$ is the height of the pixel-based monocular camera. To always track the target path approximately 0.7 [m] forward on an autonomous truck, a pixel-based ROI of 200 to 300 pixel or less was designed.

The parameters such as target RGB, position, and velocity standard deviations of particles and the number of particles are sensitive parameters and were determined through trial and error methods. To check the effect of cycle loop time according to the number of particles (200, 500, 1000, 3000, 5000, and 10,000), a total of 450 frames for the experiment were designed as total experiment frames. Figure 8a shows the cycle loop time according to the number of particles and Figure 8b shows the absolute average and root mean square (RMS) of cycle loop time. Table 3 lists the absolute average and RMS according to the number of particles. As a result, it can be observed that the cycle loop time increases as the number of particles increases. In addition, as shown in Figure 8a, when particles are dispersed by obstacles, the cycle loop time is increased because of the increase of computational load.

Figure 9 shows the yellow-tape-based configured path in the parking lot in front of Hankyong National University’s Mechanical Engineering Hall for the performance evaluation of three scenarios: elliptical, S-curved, and lane-change paths.

The next subsections highlight the performance evaluation results for the ellipse, S-curved, and lane-change scenarios. The LQR and sliding mode control (SMC)-based evaluation and comparison results are provided.

### 3.1. Path Tracking Scenario: Ellipse Path Tracking

The ellipse path tracking scenario had two 5 [m] straight sections and two semicircles with a diameter of 4 [m], as shown in Figure 10. The total driving distance was approximately 22.5 [m], and the results of driving 1 lap are shown by installing obstacles 3 times in a random path during driving.

Figure 11 shows a representative 8 frames among the total 523 frames saved in the real experiment. The red dots are generated particles, the black dots are clustered points, and the yellow bands are target paths installed by the authors. In Figure 11a–c,e–g, it can be seen that the particles are gathered on the target path in a steady-state driving situation. In Figure 11d,h, it can be seen that the particles are scattered by the black obstacle.

Figure 12a shows the voltage of VR measured in real time and the converted analog signal used to derive the map-based current steering angle. Figure 12b shows the preview distance converted to the real distance and the pixel-based derived path errors. Particles in the designed ROI were generated, and it can be seen that the preview distance was maintained at approximately 0.7 [m]. In addition, because it drives in only one direction, it can be confirmed that the lateral preview and yaw angle errors increase in the positive direction in the turning sections.

Figure 13 shows the desired steering angle, current steering angle, and steering angle error derived from both the LQR and map-based scenarios. The black solid line represents the desired steering angle, the red dotted line represents the current steering angle of the autonomous trucks, and the current steering angle tracks the desired steering angle. As can be seen in Figure 13, the chattering of the current steering angle can be confirmed because of the hardware delay from generating a pulse signal to the step motor for the steering rotation flag generated based on the particle filter, and this limitation will be overcome through algorithm and hardware optimization in the future.

Figure 14 shows the error of the RGB data and the driving flag used for the driving and braking decisions. In Figure 14a, the black dots are the Euclidean norm of the RGB data of the clustered point and the target RGB data, and the sphere consisting of the red solid line is the threshold of the error of the RGB data. When three obstacles encounter the path, the error of the RGB data increases above the threshold, the driving flag signal is changed from zero to one, and the braking signal is applied to the in-wheel motor. The driving signal is a dimensionless index that has a binary number of 0 and 1.

The autonomous truck was driven for approximately 90.8 [s] as the real time taken for the ellipse path scenario, and 0.1745 [s] per sampling instance. The total length of the path of the scenario is approximately 22.5 [m], and when the total driving time is divided, it can be confirmed that the average velocity was approximately 0.248 [m/s].

### 3.2. Path Tracking Scenario: S-Curved Path Tracking

This section presents the S-curved path tracking scenario with two semicircles with a diameter of 5 [m], as shown in Figure 15. The total driving distance was approximately 15.7 [m], and the results of driving are shown by installing obstacles twice in a random path during driving.

Figure 16 shows a representative 8 frames among the total 361 frames saved in the real experiment. It can be seen that in the steady-state driving situation, particles are gathered in the path, and when a black obstacle is encountered, particles are scattered.

Figure 17 shows the VR voltage measured in real time, converted analog signal, preview distance converted to real distance, and pixel-based path errors. Particles in the designed ROI were generated, and the preview distance was maintained at approximately 0.7 [m]. As it passes through the first inflection point in the counterclockwise direction and the second inflection point in the clockwise direction, it can be seen that the lateral preview and yaw angle errors change from the positive direction to the negative direction.

In Figure 18, the steering rotation direction flag changes according to the steering angle error, and the current steering angle tracks as the desired steering angle is derived by the deflection from the positive direction to the negative direction.

Figure 19 shows the errors of the RGB and driving flags. When obstacles encounter the path, it can be seen that the RGB error increases above the threshold, the driving flag signal is changed from zero to one, and the braking signal is applied to the in-wheel motor.

The autonomous truck was driven for approximately 63.97 [s], which was the real time taken for the S-curved path scenario, and 0.1784 [s] per sampling instance. The total path of the scenario was approximately 15.7 [m], and when the total driving time was divided, it was observed that the average velocity was approximately 0.245 [m/s].

### 3.3. Path Tracking Scenario: Lane-Change

This section describes the lane-change path tracking scenario in which a lane with a width of 3.5 [m] is changed by 5 [m], as shown in Figure 20. The total driving distance was approximately 12.1 [m], and the results of driving were obtained by installing an obstacle once in a random path during driving.

Figure 21 shows a representative 8 frames among total 263 frames saved in the real experiment. It can be seen that in the steady-state driving situation, particles are gathered in the path, and when a black obstacle is encountered, particles are scattered.

Figure 22 shows the VR voltage, converted analog signal, preview distance, and path errors. As in scenarios 1 and 2, it can be seen that the preview distance is maintained at approximately 0.7 [m].

Figure 23 confirms that the desired steering angle is changed from a negative direction to a positive direction by driving clockwise to change lanes after going straight.

Figure 24 shows that when an obstacle is encountered in the path and there is no path after reaching the target point, the error of RGB increases above the threshold, the driving flag signal changes from zero to one, and the braking signal is applied to the in-wheel motor.

It was driven for approximately 45.89 [s] as the real time taken for the lane-change path scenario and 0.1767 [s] per sampling instance. The total path of the scenario is approximately 12.1 [m], and when the total driving time is divided, it can be seen that the average velocity is approximately 0.264 [m/s].

### 3.4. Comparison of LQR- and SMC-Based Path Tracking Performance

To compare the LQR-based path tracking control performance proposed in this study with another controller, a conventional SMC of a robust controller was used to conduct performance evaluation under the elliptical path tracking scenario. The desired steering angle was derived by designing a sliding surface and a cost function based on the Lyapunov function using the derived path error-based simplified error dynamics. The conventional SMC has a chattering phenomenon if an unreasonable smoothing method is applied, so a sigmoid function was applied in this study. The SMC parameters were derived using a trial and error method to ensure reasonable path tracking performance and compare to the LQR-based evaluation results [26].

Figure 25a–d show actual pictures taken from the experiment in the case of the LQR-based elliptical path tracking scenario (Appendix A). Figure 25e–h show actual pictures taken from the experiment in the case of the SMC-based elliptical path tracking scenario.

Figure 26a shows the voltage of VR measured in real time and the converted analog signal used to derive the map-based current steering angle. The black solid line represents the LQR-based results and the red dotted line represents the SMC-based results. Figure 26b shows the current steering angle of the autonomous trucks. As can be seen in Figure 26, it was confirmed that a smaller steering angle was applied to the SMC at the second curvature point in approximately 250 to 350 sampling instances. The desired steering angle can be changed by designing the magnitude of injection term, which is the design parameter.

Figure 27a shows the derived path error, and Figure 27b shows the steering angle error derived according to LQR and SMC. As can be seen in Figure 27, the preview distance of approximately 0.7 [m] was maintained, and it was confirmed that the steering angle error occurred within approximately 3°.

Figure 28a,b show the desired steering angle derived based on LQR and SMC. The black solid line represents the desired steering angle, the red dotted line represents the current steering angle. Figure 28c,d show the Gaussian distribution of the steering angle error based on the LQR and SMC. In Figure 28a,b, it can be seen that the desired steering angle of SMC is smaller than that of LQR. In Figure 28c,d, it can be seen that the steering angle error of LQR and SMC are similar.

Table 4 and Table 5 list the absolute average, standard deviation, and RMS of LQR- and SMC-based derived lateral preview error and steering angle error. In Table 4 and Table 5, when calculated by RMS value, it was confirmed that the lateral preview error of SMC compared to LQR was approximately 18% larger than that of the LQR-based angle. The RMS of the steering angle error LQR and SMC was 1%, confirming that the steering angle error was similar. SMC parameters require changing in various curvature scenarios because the desired steering angle is derived according to the boundary value of disturbance regardless of the magnitude of the path error. However, LQR has the advantage of being applicable in various scenarios if appropriate parameters are used because the desired steering angle is derived according to the magnitude of the path error. In addition, it has relatively few design parameters compared to SMC, and has the advantage of being able to compute optimal control inputs.

## 4. Summary and Conclusions

### 4.1. Summary

Recently, research and commercialization for systems with autonomous driving technology has been conducted to improve worker safety, convenience, and efficiency in various fields, such as logistics warehouses, smart farms, and smart factories. The path tracking of these systems recognizes the surrounding environment using various methods, such as artificial intelligence based on various sensors or image processing. However, these methods require a lot of training data, the initial infrastructure configuration has a high cost, and the number of required sensors may increase. Using many sensors can increase failure probability and energy consumption. Particle filters have recently used in-object tracking and path planning methods as an advantage that can be applied to nonlinear systems without mathematical models. We focused on develop a path tracking algorithm using only target RGB without image processing and a large amount of learning data using a monocular camera based on a particle filter.

In this study, we proposed a path tracking algorithm for autonomous trucks with a single steering and driving module. The target path was tracked using a monocular camera and the target RGB data based on a particle filter. Pixel-based path errors were derived by clustering the particles generated at the points most similar to the target RGB of the path in the ROI. The LQR-based desired steering angle was derived using the path tracking errors, and the map-based current steering angle was derived through voltage measurements using VR. To track the desired steering angle, the PWM and pulse signal were applied to the step motor so that the calculated current steering angle converges to the desired steering angle.

The following is a summary of the major contributions of this study:The proposed path tracking method is an attempt to develop particle-filter-based target RGB tracking using a monocular camera.An autonomous truck with a single steering and driving module was developed, and its path tracking performance was evaluated.

### 4.2. Conclusions

In this study, a constant PWM was applied to the in-wheel motor to drive the autonomous trucks at an average velocity of approximately 0.25 [m/s]. The performance evaluation was conducted based on the developed autonomous truck under four scenarios: ellipse, S-curved, and lane-change paths, and an SMC-based elliptical path. As can be seen in the performance evaluation, the chattering of the current steering angle can be confirmed by the hardware delay while generating a pulse signal to the step motor for the steering rotation flag generated based on the particle filter, and this limitation will be overcome through algorithm and hardware optimization in the future.

The design parameters of the particle filter and LQR are sensitive and were derived through trial and error. It was confirmed that the cycle loop time also increased as the number of particles increased. In addition, to compare the LQR-based path tracking control performance proposed in this study, an elliptical path tracking scenario using conventional SMC was performed. As a result, the RMS of the lateral preview error of the SMC was approximately 18% larger than that of the LQR-based scenario. The RMS of the steering angle error between LQR and SMC was 1%, confirming that the steering angle error was similar. SMC parameters require changing in various curvature scenarios because the desired steering angle is derived according to the boundary value of disturbance, regardless of the magnitude of the path error. However, LQR has the advantage of being applicable in various scenarios if appropriate parameters are used because the desired steering angle is derived according to the magnitude of the path error. In addition, it has relatively few design parameters compared to SMC, and has the advantage of being able to compute optimal control inputs. However, LQR requires a mathematical model and has limitations in that uncertainty exists in the process of converting pixels of a camera image into real distance.

Therefore, in the near future, we will focus on improving the path tracking algorithm that use only pixels from camera images by applying model-free control methods or various adaptive theories that do not require mathematical models. In addition, adding an object tracking mode indoors and outdoors using various sensors, such as GPS for outdoor driving, and IMU and lidar for obstacle detection, is considered for future work which will fuse various sensors to advance more robust and safe path tracking algorithms by recognizing the surrounding environments and evaluating the performance in path scenarios with various curvatures. Based on the algorithm proposed in this study, it is expected that a target path and the surrounding environment can be recognized in various fields using autonomous driving technologies for various purposes.

## Figures and Tables

**Figure 1 sensors-23-03650-f001:**
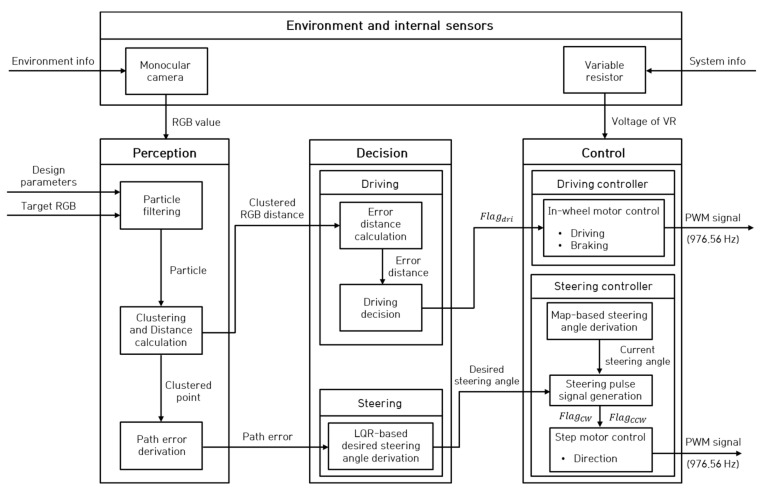
Overall block diagram of the path tracking algorithm.

**Figure 2 sensors-23-03650-f002:**
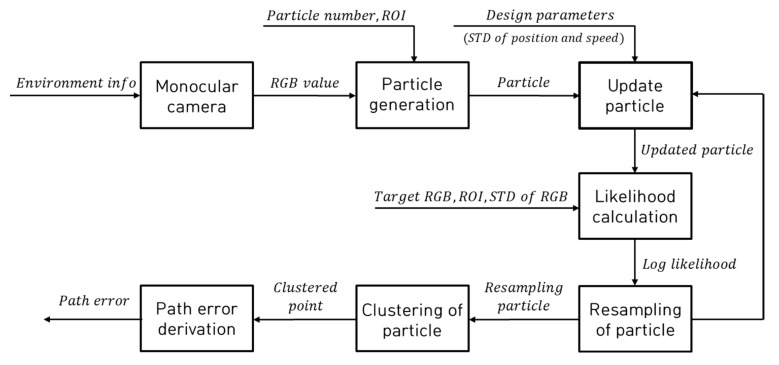
Block diagram for the particle-filter-based path tracking error derivation.

**Figure 3 sensors-23-03650-f003:**
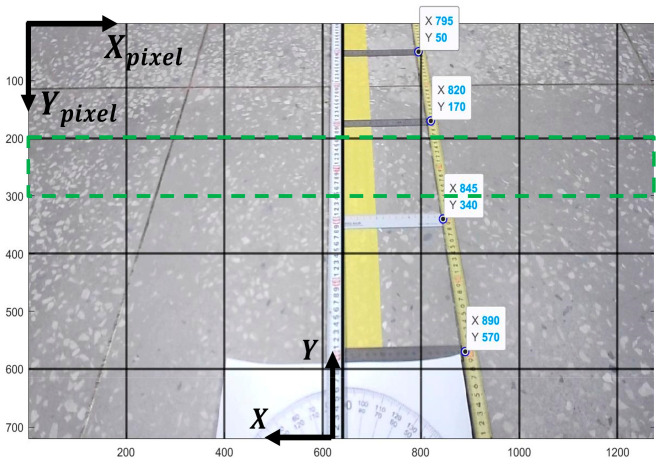
Conversion of pixel to actual distance in image.

**Figure 4 sensors-23-03650-f004:**
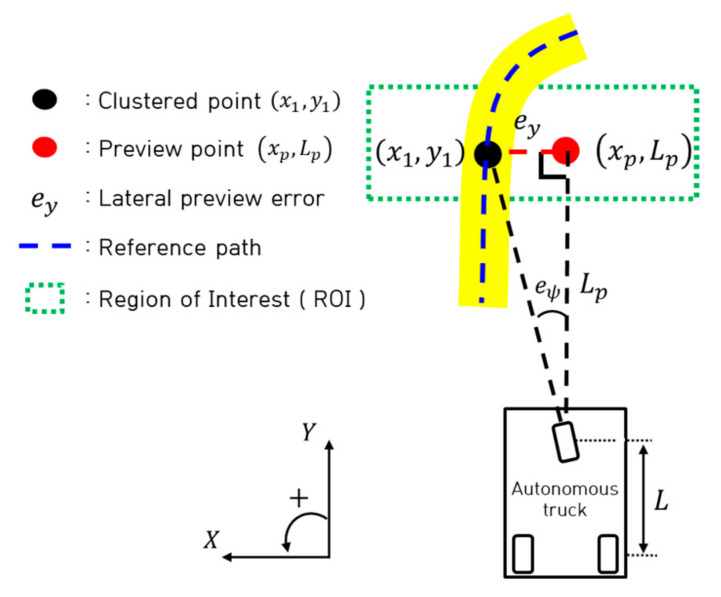
Path error derivation.

**Figure 5 sensors-23-03650-f005:**
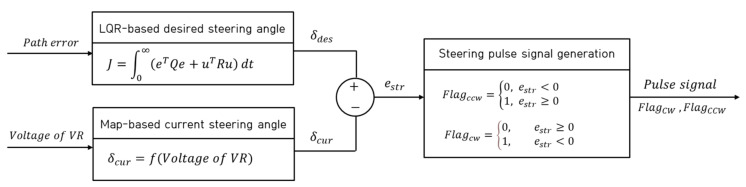
Block diagram for steering control algorithm for path tracking.

**Figure 6 sensors-23-03650-f006:**
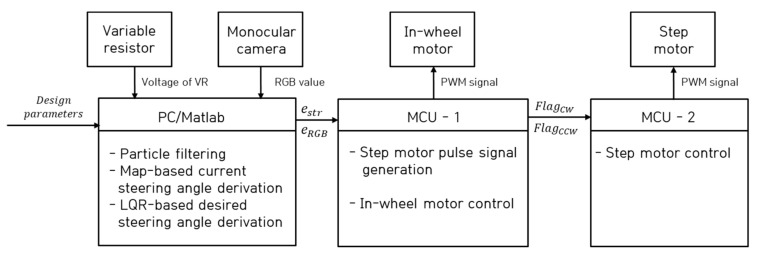
Block diagram for an autonomous truck with a single steering and driving module.

**Figure 7 sensors-23-03650-f007:**
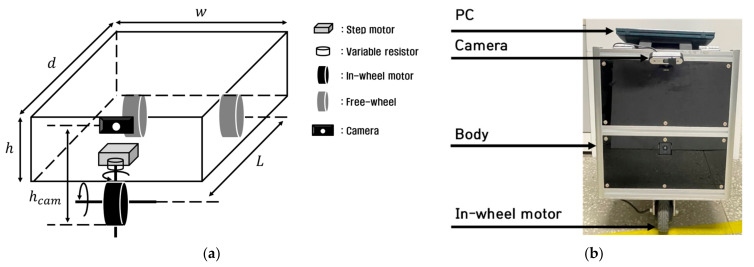
(**a**) Concept of the developed autonomous truck; (**b**) image of the developed actual autonomous truck.

**Figure 8 sensors-23-03650-f008:**
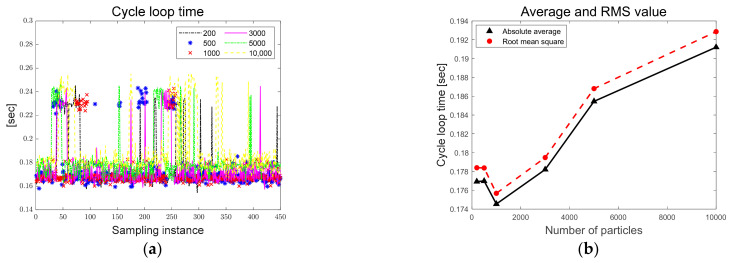
(**a**) Cycle loop time by number of particles; (**b**) average and RMS of cycle loop time.

**Figure 9 sensors-23-03650-f009:**
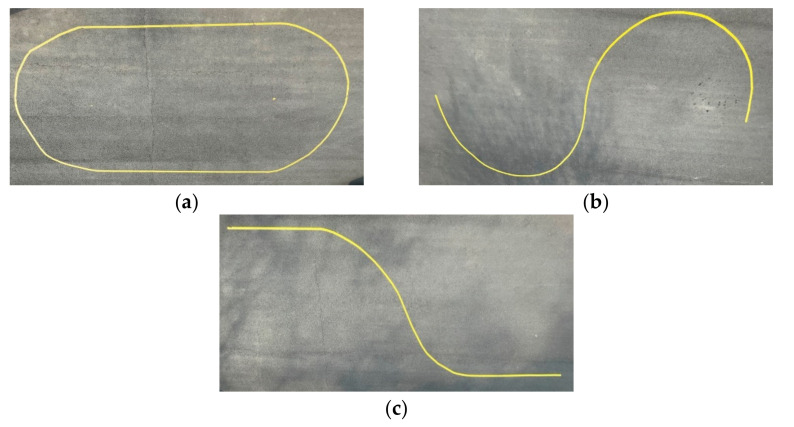
(**a**) Ellipse-path tracking scenario; (**b**) S-curved path tracking scenario; (**c**) lane-change path tracking scenario.

**Figure 10 sensors-23-03650-f010:**
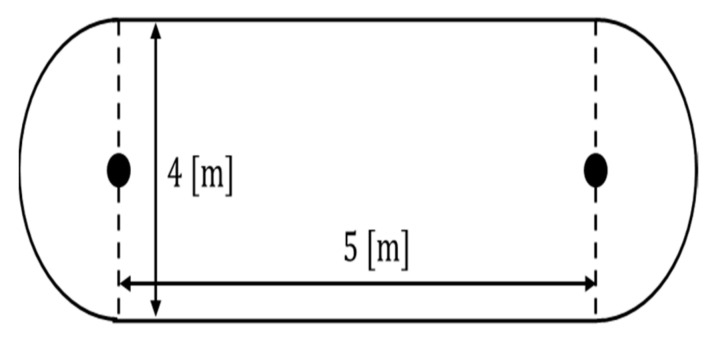
Ellipse path tracking scenario.

**Figure 11 sensors-23-03650-f011:**
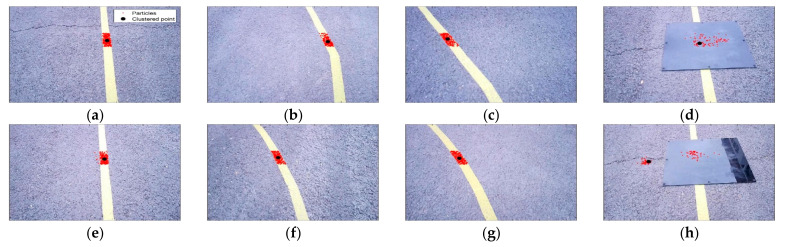
(**a**) Actual experiment image—60th frame; (**b**) actual experiment image—120th frame; (**c**) actual experiment image—180th frame; (**d**) actual experiment image—252nd frame; (**e**) actual experiment image—320th frame; (**f**) actual experiment image—380th frame; (**g**) actual experiment image—430th frame; (**h**) actual experiment image—509th frame.

**Figure 12 sensors-23-03650-f012:**
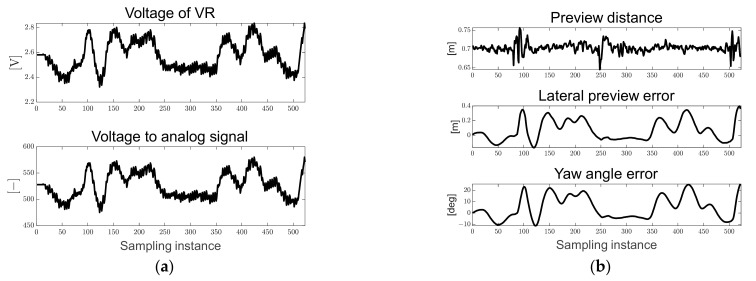
Ellipse path tracking scenario results; (**a**) voltage of VR and analog signal for map-based calculation; (**b**) path errors: preview distance, lateral preview and yaw angle errors.

**Figure 13 sensors-23-03650-f013:**
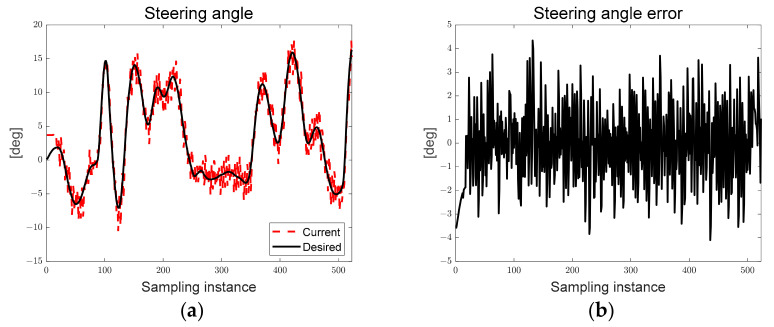
Ellipse path tracking scenario results; (**a**) desired and current steering angles; (**b**) steering angle error.

**Figure 14 sensors-23-03650-f014:**
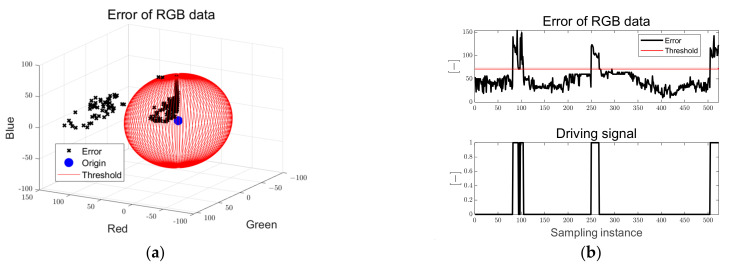
Ellipse path tracking scenario results; (**a**) error of RGB data; (**b**) driving flag signal.

**Figure 15 sensors-23-03650-f015:**
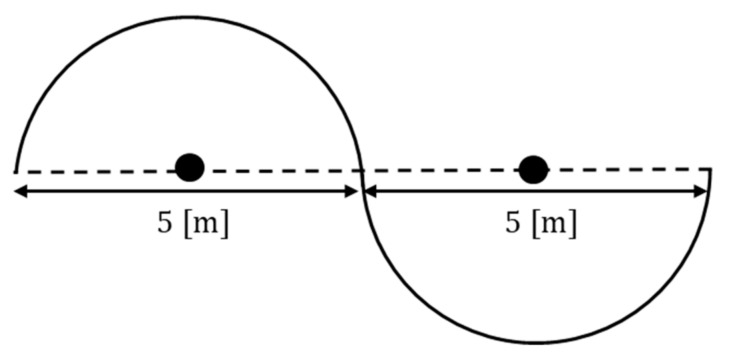
S-curved path tracking scenario.

**Figure 16 sensors-23-03650-f016:**
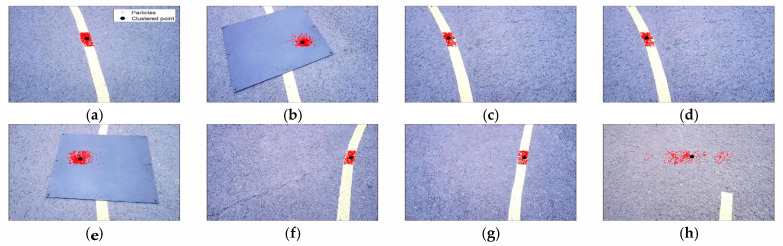
(**a**) Actual experiment image—22nd frame; (**b**) actual experiment image—58th frame; (**c**) actual experiment image—100th frame; (**d**) actual experiment image—150th frame; (**e**) actual experiment image—178th frame; (**f**) actual experiment image—230th frame; (**g**) actual experiment image –300th frame; (**h**) actual experiment image—360th frame.

**Figure 17 sensors-23-03650-f017:**
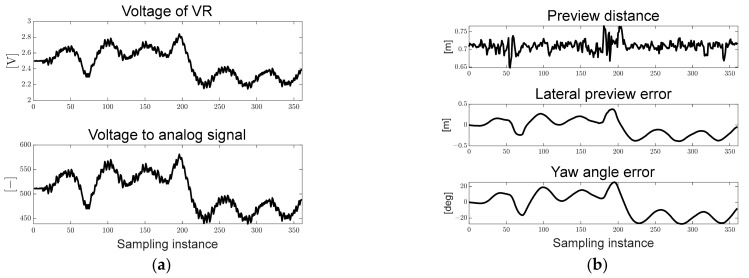
S-curved path tracking scenario results; (**a**) voltage of VR and analog signal for map-based calculation; (**b**) path errors: preview distance, lateral preview and yaw angle errors.

**Figure 18 sensors-23-03650-f018:**
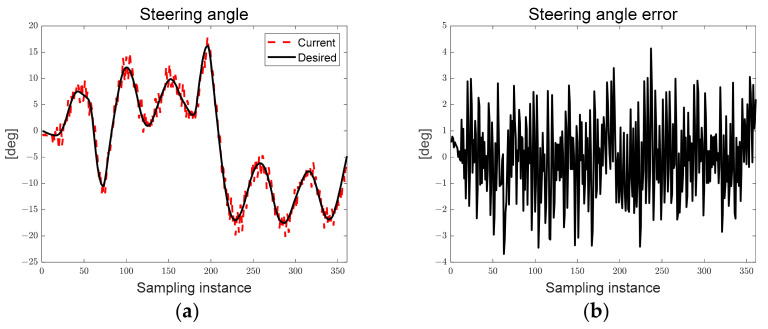
S-curved path tracking scenario results; (**a**) desired and current steering angles; (**b**) steering angle error.

**Figure 19 sensors-23-03650-f019:**
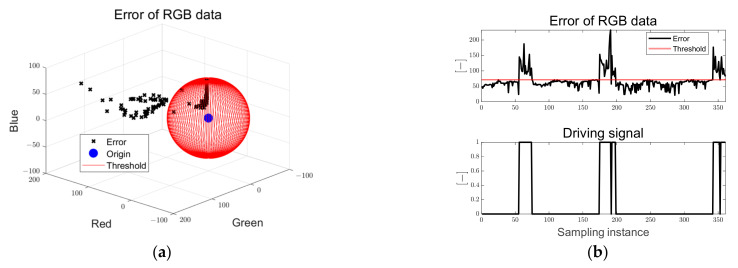
S-curved path tracking scenario results; (**a**) error of RGB data; (**b**) driving flag signal.

**Figure 20 sensors-23-03650-f020:**
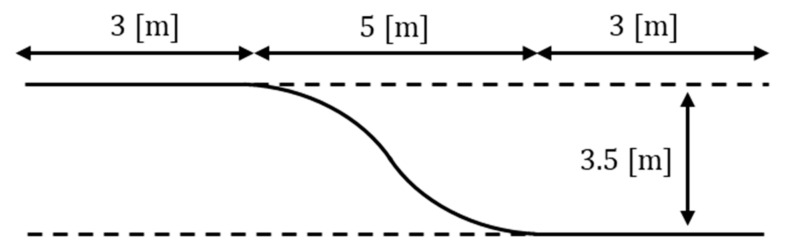
Lane-change path tracking scenario.

**Figure 21 sensors-23-03650-f021:**
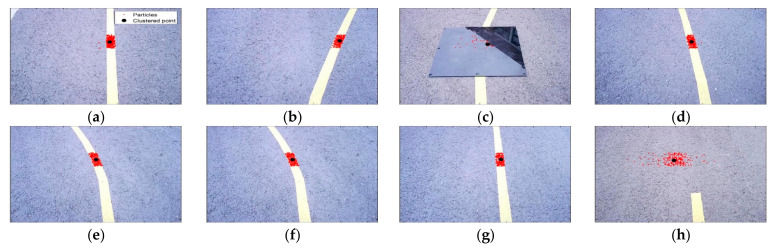
(**a**) Actual experiment image—30th frame; (**b**) actual experiment image—70th frame; (**c**) actual experiment image—110th frame; (**d**) actual experiment image—150th frame; (**e**) actual experiment image—180th frame; (**f**) actual experiment image—200th frame; (**g**) actual experiment image—230th frame; (**h**) actual experiment image—260th frame.

**Figure 22 sensors-23-03650-f022:**
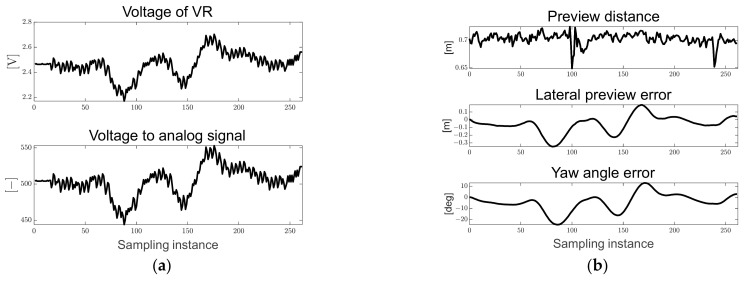
Lane-change path tracking scenario results; (**a**) voltage of VR and analog signal for map-based calculation; (**b**) path errors: preview distance, lateral preview and yaw angle errors.

**Figure 23 sensors-23-03650-f023:**
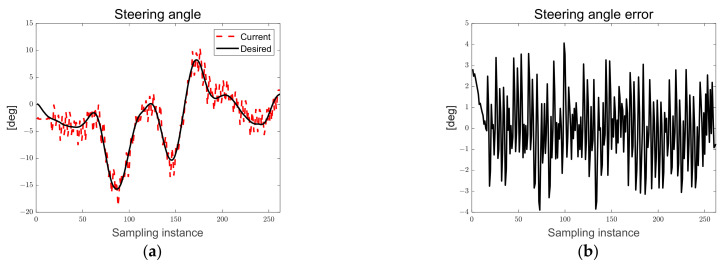
Lane-change path tracking scenario results; (**a**) desired and current steering angles; (**b**) steering angle error.

**Figure 24 sensors-23-03650-f024:**
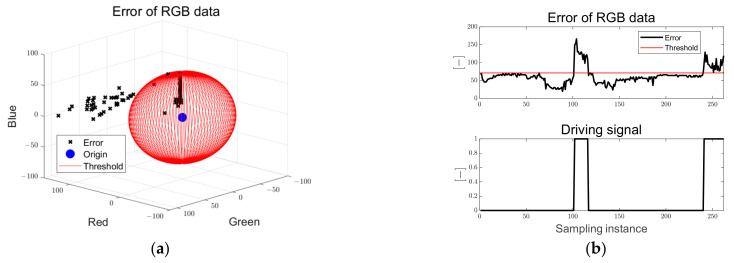
Lane-change path tracking scenario results; (**a**) error of RGB data; (**b**) driving flag signal.

**Figure 25 sensors-23-03650-f025:**
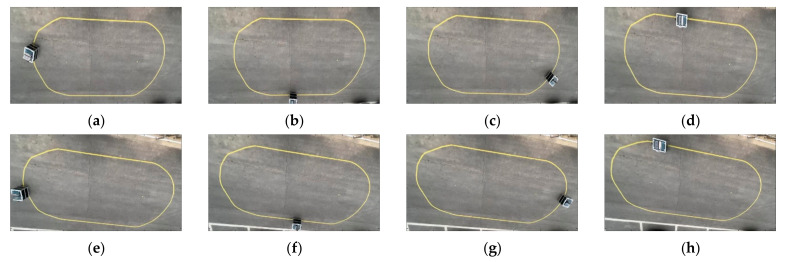
Ellipse path tracking scenario results; (**a**) LQR-based actual experiment image—600th frame; (**b**) LQR-based actual experiment image—1200th frame; (**c**) LQR-based actual experiment image—1800th frame; (**d**) LQR-based actual experiment image—2800th frame; (**e**) SMC-based actual experiment image—600th frame; (**f**) SMC-based actual experiment image—1200th frame; (**g**) SMC-based actual experiment image—1800th frame; (**h**) SMC-based actual experiment image—2600th frame.

**Figure 26 sensors-23-03650-f026:**
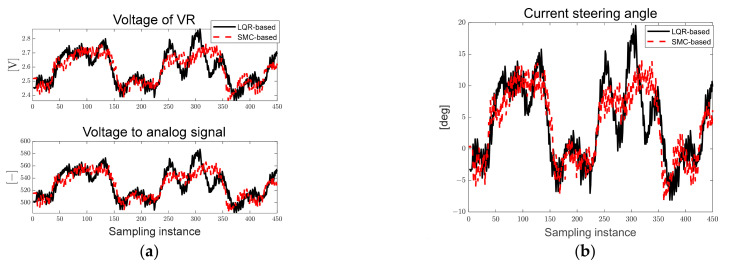
Ellipse path tracking scenario results; (**a**) voltage of VR and analog signal for map-based calculation; (**b**) current steering angle.

**Figure 27 sensors-23-03650-f027:**
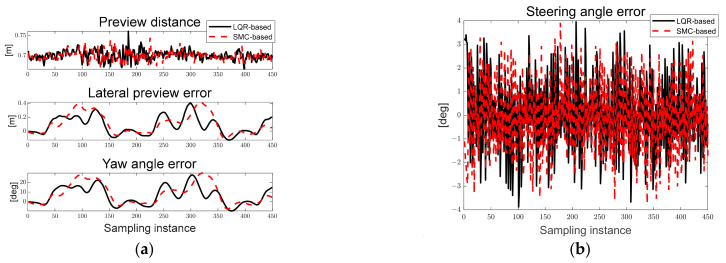
Ellipse path tracking scenario results; (**a**) path errors: preview distance, lateral preview and yaw angle errors; (**b**) steering angle error.

**Figure 28 sensors-23-03650-f028:**
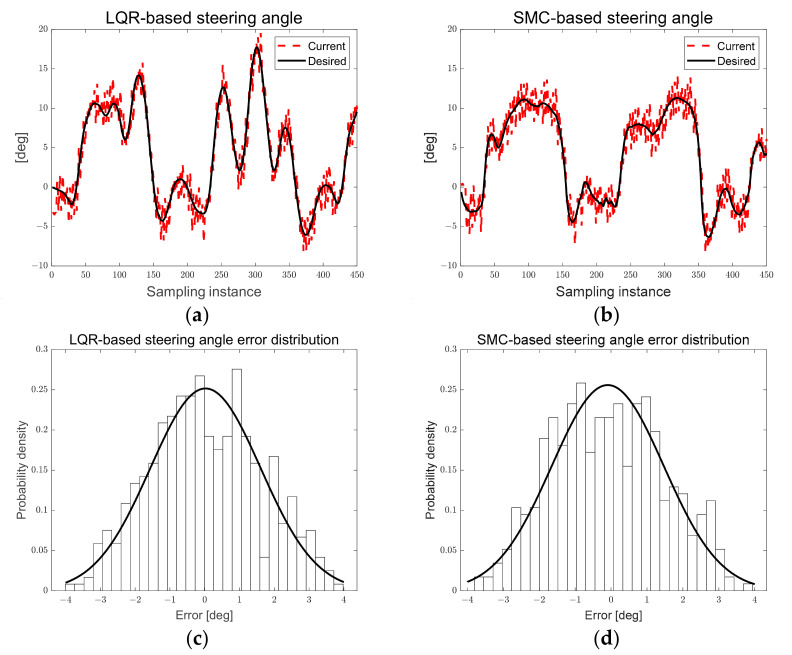
(**a**) LQR-based steering angle; (**b**) SMC-based steering angle; (**c**) LQR-based steering angle error distribution; (**d**) SMC-based steering angle error distribution.

**Table 1 sensors-23-03650-t001:** Autonomous truck and camera specifications.

Parameter	Unit	Value
Mass (m)	kg	38.6
Height of truck (*h*)	m	0.45
Depth of truck (*d*)	m	0.55
Width of truck (*w*)	m	0.45
Wheel base (*L*)	m	0.5
Wheel tread ($t_{w}$)	m	0.375
Height of camera ($h_{cam}$)	m	0.6
Angle of camera ($\theta_{cam}$)	deg	−48
Resolution of camera	pixel	1280 × 720

**Table 2 sensors-23-03650-t002:** Design parameters of the particle filter and LQR.

Parameter	Unit	Value
Standard deviation of RGB	-	20
Standard deviation of position	-	20
Standard deviation of velocity	-	15
The number of particles	-	1000
Target RGB ($r_{t},g_{t},b_{t}$)	-	(254, 252, 184)
Error of RGB data threshold ($d_{th}$)	-	72
ROI	-	$200<h_{pixel}\leq 300$
Error states weighting matrix	-	$\left[\begin{array}{cc} 100 & 0\\ 0 & 10 \end{array}\right]$
Input weighting matrix	-	650
LQR gain	-	${\left[\begin{array}{cc} 0.3922 & 0.6070 \end{array}\right]}^{T}$

**Table 3 sensors-23-03650-t003:** Absolute average and RMS of cycle loop time.

Number of Particles	Absolute Average	RMS
200	0.1769 [s]	0.1784 [s]
500	0.1770 [s]	0.1784 [s]
1000	0.1745 [s]	0.1757 [s]
3000	0.1782 [s]	0.1795 [s]
5000	0.1855 [s]	0.1868 [s]
10,000	0.1912 [s]	0.1929 [s]

**Table 4 sensors-23-03650-t004:** LQR-based ellipse path tracking error.

Description	Lateral Preview Error	Steering Angle Error
Absolute average	0.1159 [m]	1.3 [deg]
RMS	0.1505 [m]	1.5857 [deg]
Standard deviation	0.1258 [m]	1.5873 [deg]

**Table 5 sensors-23-03650-t005:** SMC-based ellipse path tracking error.

Description	Lateral Preview Error	Steering Angle Error
Absolute average	0.1287 [m]	1.3 [deg]
RMS	0.1776 [m]	1.5619 [deg]
Standard deviation	0.1439 [m]	1.561 [deg]

## Data Availability

Not applicable.

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
