# Peer review of "Development of a Particle Filter-Based Path Tracking Algorithm of Autonomous Trucks with a Single Steering and Driving Module Using a Monocular Camera"

_sensors, 2023, doi:10.3390/s23073650_

Round 1
Reviewer 1 Report
Figure 1 should be described more precisely. The reader would like to understand how the components of the system interact with each other. This is a very important drawing for understanding the idea of the proposed system.
The paper contains too many drawings, as many as 32 - including many multiple drawings. Understandably, the authors wanted to give credibility to the research done. However, the paper should have included only the most important ones, the most representative pieces. With such a large number of drawings, there was not enough space for their interpretation.
Each drawing should be described - the authors should describe what it shows, why the graphs are shaped the way they are - whether this is good or bad. The article should include an evaluation of the results.
There are few conclusions in the paper - the conclusions can be the interpretation of the results that are in the figures.
The final conclusions should not necessarily include a summary of the research or a description of the solution. They lack reference to the solutions given in the literature - in what is the solution presented better, where is it comparable or perhaps where is it weaker than others have done?
The language of the paper is understandable, although there are excessively long sentences that can be divided into two sentences. Do algorithm and diagram mean the same thing - in the work they are used interchangeably, or they are combined which is incomprehensible (e.g. Figure 5. Block diagram for steering control algorithm for path tracking. ) - or Block diagram or Control algorithm.
The title of the subsection "1.2. Summary of the proposed control algorithm and major contributions" at the beginning of the paper is inappropriate. The summary is included after the results are presented.
The work substantively is of a high level. From its content you can see that the authors have put a lot of work, and they understand the issue and have done the research independently.
The weakest side of the evaluated work is its editing, and in this area the work needs improvement:
- standardize references to equations - for example, page 8 in the work is "Eqs. (19)" vs. "Equation (21)",
- Figure 23 is drawn with a line that is too thick and the characters in the figure are in too large a font,
- align single drawings to the edges of the text according to Sensors template (e.g. 8, 9, 15, 16, ...),
- sign drawings (8, 10, 11, 12, ..., 17 ...) multiple according to Sensors template (Figure 2. template),
- describe the axes of the graphs in units - e.g. Driving signal in Figure 21.
In the final conclusions, include a summary of the results obtained.
Author Response
Thank you for your valuable comments on our submitted manuscript. The response file is attached and the manuscript revised by authors based on reviewers' valuable comments is also uploaded. Please, check the response file and revised manuscript. Thank you very much.

Reviewer 2 Report
Summary
This study provided a particle filter-based route tracking system for autonomous vehicles employing a monocular camera and a single steering and driving module. The program estimates path tracking errors and produces a desirable steering angle based on a linear quadratic regulator, which is utilized to steer and control the autonomous vehicles. The suggested approach has been demonstrated to be successful in tracking the goal route under various conditions, and future work will concentrate on enhancing the algorithm utilizing multiple sensor types.
General concept comments
·        The abstract doesn’t mention the specific problem to solve it is only stated that the reconfiguration of the truck is expensive.
·        It would be perfect to add a specific KPI or a metric performance of the algorithm in the abstract.
Specific comments:
·        What are the main differences between the performance of the proposed algorithm, with others like: Kalman-filter, Model Predictive Control, Fuzzy logic or the pure pursuit algorithm?
·        Comparing to the previous control algorithms what are the main improvements?
Major points
The control algorithm and the results are very satisfying and well presented?
The algorithm used is the Sequential Monte Carlo methods or as stated the particle filter based algorithm is very powerful for object tracking, localization Signal processing and also control, However the authors must state the following problems and how to solve them, this will give a very competitive approach to overcome the latest published control solutions:
·        The computations are intensive this can lead for long time to take decisions comparing to others, can the authors add a KPI or a metric that mention this problem?
·        The algorithm is very sensitive to tuning parameters therefore it can be hard to implement and time consuming.
·        Also how the authors solves the particle degeneracy problem where particles with high weight can dominate the processing?
·        And how will the algorithm integrate more Sensors for example: GPS, IMU, Camera and Lidar in parallel ?
Minor points
·        Figures 20, 30, 31 and 32 are not addressed and explained in the text.
·        Introduction is well presented however the research gap must be stated.
·        Can the author publish a video or a data of the results after accepting, because it will boost the credibility of the research and also increase the visibility for high citations?
Author Response

(The authors gave the same response as above.)

Reviewer 3 Report
This paper develops a particle filter-based path tracking algorithm of autonomous trucks with a single steering and driving module using a monocular camera. In general, the paper is interesting and some comments are listed as follows:
Â
1.     The description of the abstract should be standardized, and the information of the whole study should be summarized in the order of background, purpose, research methods and conclusions.
2.     The introduction appropriately reduces the description of literatures, and the logicality and hierarchy of the upper and lower parts need to be enhanced. Explain why the literature should be mentioned instead of simply summarizing the work in a whole paragraph.
3.     Some sentences in the paper are too long. Pay attention to the standardization of language.
4.     Mathematical expressions should be standardized. For example, whether equations (21)-(23) can be in subsection form.
5.     The shooting of pictures should be more professional. For example, the reflection of human shadow in Figure 7(b) is not allowed. The three scene maps in Figure 8 try to adopt the form of top view.
6.     The distance of the tracking experiment snapshots in the three environments of ellipse, S-shape and lane change path is somewhat close, and the shooting distance should be appropriately increased.
7.     In the paper, the experimental data should be appropriately added to make the research more stereoscopic.
8.     The conclusion is a summary of the whole work. The description should be refined. The conclusion and outlook can be divided into two paragraphs.
9.     The format of references should be consistent. For example, the uppercase and lowercase of the article name and the DOI number of each article.
10. Neural network-based path planning methods should be discussed in the literature review part, e.g.,  A Neural Network-Based Navigation Approach for Autonomous Mobile Robot Systems, Applied Sciences 12 (15), 7796.
Â
Â
Author Response

(The authors gave the same response as above.)

Round 2
Reviewer 3 Report
The authors have addressed all my comments.